# E-cadherin Expression in Canine Gastric Carcinomas: Association with Clinicopathological Parameters

**DOI:** 10.3390/vetsci9040172

**Published:** 2022-04-01

**Authors:** Ana R. Flores, Alexandra Rêma, João R. Mesquita, Marian Taulescu, Fernanda Seixas, Fátima Gärtner, Irina Amorim

**Affiliations:** 1ICBAS—School of Medicine and Biomedical Sciences, University of Porto, 4050-313 Porto, Portugal; anaruteflores@gmail.com (A.R.F.); alexandra.rema@gmail.com (A.R.); jrmesquita@icbas.up.pt (J.R.M.); iamorim@ipatimup.pt (I.A.); 2Institute of Pathology and Molecular Immunology, University of Porto (IPATIMUP), 4200-465 Porto, Portugal; fgartner@ipatimup.pt; 3CECAV—Veterinary and Animal Research Center, University pf Trás-os-Montes and Alto Douro (UTAD), 5001-801 Vila Real, Portugal; fseixas@utad.pt; 4Associate Laboratory for Animal and Veterinary Sciences (AL4AnimalS), 5001-801 Vila Real, Portugal; 5Epidemiology Research Unit (EPIUnit), Instituto de Saúde Pública, Universidade do Porto (ISPUP), 4050-600 Porto, Portugal; 6Department of Pathology, Faculty of Veterinary Medicine, University of Agricultural Sciences and Veterinary Medicine, 400372 Cluj-Napoca, Romania; 7Synevovet Laboratory, 81 Pache Protopopescu, 021408 Bucharest, Romania; 8i3S—Instituto de Investigação e Inovação em Saúde, Universidade do Porto, 4200-135 Porto, Portugal

**Keywords:** dog, gastric carcinoma, stomach, E-cadherin

## Abstract

E-cadherin (E-cad) is a cell-adhesion molecule known for its tumor-invasion suppressor function. E-cad expression was examined immunohistochemically in a series of canine tissue samples, including normal gastric mucosa (NGM; n = 3), gastric carcinomas (GC; n = 33), adjacent non-neoplastic mucosa (NNM; n = 32), neoplastic emboli (n = 16) and metastatic lesions (n = 9). The relationship between E-cad expression and clinicopathological features were investigated. In NGM, epithelial cells showed strong latero-lateral membranous expression of E-cad, and this pattern was considered normal. The membranous staining was preserved in all specimens of NNM (100%), whereas abnormal E-cad expression was found in 87.9% of the GCs. A marked difference in E-cad expression was observed between normal and malignant tissues (*p* < 0.0002). Abnormal E-cad expression was significantly more frequent in poorly/undifferentiated carcinomas (96%) and diffuse (95%) and indeterminate carcinomas (100%) than in well-differentiated/intestinal ones (62.5%; *p* = 0.0115 and *p* = 0.0392, respectively). There was significant association between abnormal E-cad expression and the depth of invasion (*p* = 0.0117), and the presence neoplastic emboli (*p* = 0.0194). No statistically significant differences in E-cad expression were observed concerning tumor location, histological type according to WHO classification, and presence of metastatic lesions. Therefore, deregulation of E-cad expression may play a role in canine gastric carcinogenesis and in tumor progression; moreover, it might be a prognostic tool for canine gastric cancer.

## 1. Introduction

Gastric cancer is one of the most common and deadly cancers worldwide [1]. In canine species, gastric tumors are rare, representing less than 1% of all reported neoplasms in dogs. However, carcinomas are the most frequent neoplasms, comprising 50–90% of all canine gastric malignancies [2].

The median age of dogs with GC ranges from 8 to 10 years, but occasional cases have been reported in dogs younger than 5 years [3]. As in man, GC occurs more frequently in males and a higher incidence of GC has been reported in Belgian Shepherd, Rough Collie, Staffordshire Bull Terrier, Chow-Chow, and standard Poodle [2]. There are also case reports of gastric cancer in association with Ménétrier’s disease in West Highland White Terrier [4] and a family of Cairn Terriers [5]. The high incidence in individual breeds has emphasized the idea that gastric cancer may be a heritable disease in dogs [6].

In dogs, the most common associated clinical signs are vomiting, anorexia, weight loss and lethargy, but many others (such as ptyalism, gagging, retching, apathy, and cachexia, and occasionally melena and abdominal pain) can be present [3,7,8,9].

In humans, early diagnosis of gastric cancer is an important factor for long-term survival, and endoscopy with gastric mucosal biopsy collection is the most valuable diagnostic tool [10]. In recent years, gastric cancer in dogs has been diagnosed with increasing frequency, probably due to the use of more accurate diagnostic techniques such as upper digestive endoscopy [2]. Despite this, the early stages of the disease are often asymptomatic and the late onset of clinical symptoms, at a time point when tumor is already metastasized, awards GC canine patients a poor prognosis and limited treatment options [2,3,11]. Treatment involves surgical resection that is often complicated by diffuse mural infiltration, distant metastases, carcinomatosis, and frequently a debilitated patient [2,11,12].

According to the World Health Organization (WHO) classification for domestic animals GCs is subdivided into five major histological subtypes: papillary, tubular, mucinous, signet ring cell, and undifferentiated carcinoma [13]. However, recent studies demonstrated that some canine gastric neoplastic lesions fit specific histological types only described in the human WHO classification, such as poorly cohesive and mixed carcinomas [2,6]. The WHO classification schemes are useful in the recognition of the morphological patterns, but offer little prognostic significance [13,14]. An alternative approach is the human Lauren classification, which allows GCs classification into intestinal, diffuse, and indeterminate types, and has been used to investigate the prognosis in human GCs [15,16]. To our knowledge, a correlation between histological type according to Lauren and clinical behavior and/or prognosis has not yet been demonstrated in canine GCs, although this scheme has been successfully adapted to dogs [7,9,17,18,19].

Cadherins are a family of cell-surface glycoproteins involved in calcium-dependent homotypic cell–cell adhesion that play critical roles during embryogenesis and in the maintenance of normal adult tissue architecture [20,21]. E-cadherin, a 120 kDa protein encoded by the *CDH1* gene, belongs to the classical cadherins subfamily and it is expressed in epithelial cells [21,22,23]. It has an extracellular domain (N terminal) that binds with high specificity to similar domains on adjacent cells, and an intracellular domain (C terminal) that binds to cytoskeleton proteins though catenins (ß-, α-, and γ-catenins) [21,24].

Evidence indicates that alterations in the adhesion properties between cells give them an invasive and migratory phenotype. Indeed, changes in expression or function of cell adhesion molecules, such as E-cad, have been implicated in tumor progression of most carcinomas, leading to tissue disorder, cellular de-differentiation, increased invasiveness of tumor cells, and ultimately metastasis. Therefore, it is well accepted that E-cad plays an important role as an invasion suppressor gene/protein [25,26].

In human gastric cancer, reduction/loss or abnormal expression of E-cad has been associated with poorly/undifferentiated carcinomas and diffuse-type carcinomas [27,28,29]. Moreover, altered expression of E-cad has been correlated with pathological parameters of tumor aggressiveness, namely high tumor grade and presence of metastasis, and/or poor survival rates [29,30,31]. To our knowledge, such a relationship has not yet been demonstrated in canine gastric carcinomas, although reduction/loss of E-cad expression has been previously reported in these lesions [32,33].

The aim of the present study was to investigate the role of E-cad in canine gastric carcinogenesis. In this regard, an immunohistochemical evaluation of E-cad was performed on a series of normal and malignant neoplastic canine gastric tissues and the protein immunoexpression was assessed for its association with clinicopathological features of the tumors.

## 2. Materials and Methods

### 2.1. Sample Selection

Thirty-three canine GCs were selected from the archives of the Laboratory of Veterinary Pathology, ICBAS-UP (Portugal), where they were received between 2004 and 2021. These specimens were collected during endoscopic procedures, surgery, or postmortem examination. Full-thickness biopsies were performed in 23 cases; partial biopsies including mucosa, submucosa, and tunica muscularis were carried out in seven cases; in the remaining three cases partial biopsies only included the mucosa and submucosal layers. In addition, as positive controls, samples of normal canine body and antral gastric mucosa were collected during necropsy of three dogs aged 3–10 years which died of causes not related to gastrointestinal disease.

The present study was approved by Animal Welfare Organization (ORBEA) of the ICBAS-UP, authorization N° 201/2017. All the examined samples were collected for diagnostic purposes as part of routine standard care, based on the best clinical judgement of their attending practitioners, and the investigators had no influence on the execution of any clinical procedures. Informed consent on the collection of tissue samples and the clinical follow up was obtained from patients’ owners.

Epidemiological data (breed, age, sex) of the dogs diagnosed with GC were collected from the histopathological request’s forms.

### 2.2. Histological Classification

Tissues were fixed in 10% buffered formalin and paraffin embedded. Serial consecutive 2 μm-sections were cut and processed for routine staining (hematoxylin and eosin, HE) and for immunohistochemistry study.

Sections were independently examined by three veterinary pathologists (MT, FG and IA).

Normal gastric tissues were considered as such according to previously proposed criteria [34] and were negative for the presence of *Helicobacter* spp. (confirmed by modified Giemsa stain and anti-*H. pylori* immunohistochemistry using a polyclonal antibody [RBK012; Zytomed, Berlin, Germany, diluted 1:200]).

GC cases were classified following the diagnostic criteria of human WHO [35] and Lauren histological classification schemes [36], since the one currently used for domestic animals does not comprise all the histological tumor subtypes present in this study. The human WHO classification divided GCs into tubular, papillary, mucinous, signet ring cell, poorly cohesive, and mixed carcinomas. The Lauren´s scheme divided GCs into the following categories: intestinal type, when they contained well-polarized epithelial cells organized in tubular structures; diffuse type, when they presented anaplastic cells which develop poorly defined tubular patterns with highly infiltrative growth; and indeterminate types, when they contained equal proportions of intestinal and diffuse characteristics [36]. Malignant tumors were further grouped into well-differentiated when neoplastic cells formed tubules and papillary structures, and poorly/undifferentiated when the neoplastic cells failed to form distinct structures. Additionally, the clinicopathological characteristics evaluated included tumor location, depth of tumor invasion, presence of neoplastic emboli and metastases. Tumor location was microscopically confirmed in each case. Although the depth of tumor invasion of the gastric wall was recorded in every case as the deepest layer invaded: mucosa, submucosa, tunica muscularis and serosa, only cases that included all layers of the gastric wall (full-thickness biopsies) were considered for statistical purposes. The presence of neoplastic emboli was considered whenever tumor cells were observed invading through a vessel wall and endothelium or when neoplastic cells were observed within a space lined by lymphatic or blood vascular endothelium [37].

### 2.3. Immunohistochemistry (IHC)

For the immunohistochemical study, sections were deparaffinized, hydrated, and antigen retrieval was performed in a microwave oven at 750 W with 0.5 mL Extran (Merck, Frankfurt, Germany) in 1000 mL distilled water for 10 minutes (min) after boiling. Slides were cooled for 10 min at room temperature and rinsed twice in triphosphate buffered saline (TBS, Cell marque, Merck, Darmstadt, Germany) for 5 min. The NovolinkTM Max-Polymer detection system (Novocastra, Newcastle, UK) was used for visualization according to the manufacturer´s instructions. Slides were incubated overnight at 4 °C with a monoclonal mouse anti-human E-cadherin (clone 4A2C7; diluted 1:50; Zymed, San Francisco, CA, USA). Sections were rinsed with TBS between each step of the procedure. Color was developed with 3,3-diamino-benzidine (DAB, Sigma, St. Louis, MO, USA) and sections were then counterstained with hematoxylin, dehydrated and mounted.

For negative controls, the primary antibody was replaced by an antibody of the same immunoglobulin isotype at the same concentration. Sections of normal canine gastric mucosa, known to express E-cad, were used as positive control tissue. When available, normal canine gastric mucosa present at the periphery of each tumor was also used as internal positive control. The fibroblasts and lymphocytes in normal samples and inside tumor areas were considered as internal negative controls. All samples, together with the appropriate positive and negative controls, were stained simultaneously.

### 2.4. Immunohisochemistry Evaluation of E-cadherin

E-cadherin expression was independently evaluated by three observers (ARF, FG and IA). When there was a disagreement, a consensual diagnosis was achieved through simultaneous observation using a multi-head microscope. The expression of E-cad in neoplastic cells was compared with that of epithelial cells in the normal and NNM. Staining was scored in a semiquantitative fashion from 0 to 3, according to the evaluation criteria described by Jawhari et al. [38]: 0, absence of staining; 1, diffuse cytoplasmic staining; 2, heterogeneous staining (e.g., when tumors were composed of both normal and abnormally staining areas); and 3, normal membranous pattern of staining. Because staining patterns often varied within the same tumor, particularly according with the lesion differentiation degree, the score was based on the dominant pattern. Cases displaying more than one pattern were classified as heterogeneous whenever they were present in more than 10% of the tumor area. For the ease of data analysis, all tumors with loss of normal membranous pattern of E-cad staining were classified as abnormal (e.g., scores of 0, 1, and 2). Tumors with a normal membranous pattern (score 3) were classified as having normal expression of E-cad.

### 2.5. Statistical Analysis

All statistical analyses were carried out using GraphPad Prism 5 (GraphPad Software Inc., La Jolla, CA, USA). The relationship between the histological type according to Lauren classification and between the tumor differentiation and age, sex and weight of the dogs was assessed by chi-square test. The association between E-cad expression and histological type, tumor differentiation and various clinicopathological parameters of the tumors was evaluated using chi-square test. Differences were considered statistically significant at values of *p* < 0.05.

## 3. Results

### 3.1. Clinicopathological Data

The available epidemiological data (breed, age, sex), characteristics of the tumors and immunohistochemical results are summarized in Table 1. This study included seven crossbreeds (21.2%), four Chow-Chows (12.1%), two Collies, two Poodles, two Labrador Retrievers, two Golden Retrievers, two Siberian Huskies, and 12 dogs of others breeds. Five dogs were considered small breed (≤10 kg; 15.2%), seven medium breed (11–25 kg; 21.2%), and 19 large breed (26–45 kg; 57.6%), in the remaining two dogs these data were not available. The age of the dogs at the time of diagnosis ranged from 5 to 14 years, with a mean age of 10.1 years ± 2.6. The male to female ratio was 1.36:1. The existence of clinical signs of gastric disease was mentioned in 18 cases (54.5%) and in one case it was an accidental finding; in the remaining 14 cases these data were not available. The most consistent signs were vomiting (n = 15; 45.5%) and weight loss (n = 12; 36.4%). Other clinical signs included anorexia (n = 6), anemia (n = 6), melena (n = 5), hematemesis (n = 5), lethargy (n = 4), hyporexia (n = 3), hematochezia (n = 2), drooling (n = 1), and diarrhea (n = 1). Tumors were located in the antral region (n = 16; 48.5%), gastric body (n = 13; 39.4%), and in both the body and antral regions (n = 3; 9.1%). In the remaining case (3.0%) these data were not available. According to the WHO criteria the gastric carcinomas included in this study were histologically classified as: papillary (n = 1; 3.0%), mucinous (n = 2; 6.1%), tubular (n = 7; 21.2%), signet ring cell (n = 11; 33.3%), poorly cohesive (n = 7; 21.2%), and mixed (n = 5; 15.2%).

Based on Lauren classification, 8 tumors were of intestinal type (24.2%), 20 were of diffuse type (60.6%) and the remaining 5 cases were of indeterminate type (15.2%; Figure 1A–E).

Twenty-five of the 33 GC cases were poorly/undifferentiated, and the remaining eight cases were well-differentiated. There was a significant association between tumor differentiation and the sex of the animals, as poorly/undifferentiated carcinomas were more frequently detected in male (17/19, 89.5%) than in female dogs (8/14, 57.1%; *p* = 0.0322; Table 2). No significant differences were observed between tumor differentiation and age or weight of the dogs or between the histological type of tumors, according to Lauren classification, and sex, age, or weight of the dogs (Table 2).

Regarding the depth of tumor invasion, 16 cases (48.5%) invaded the tunica muscularis, 12 cases (36.4%) the serosal layer, 4 cases (12.1%) were limited to mucosa, and 1 case (3.0%) to the submucosal layer of the gastric wall. Most cases had neoplastic emboli (n = 18, 54.5%), which were usually observed within lymphatic vessels. For IHC evaluation only 16 cases with neoplastic emboli were available; in the remaining two cases it was not possible to evaluate due to tissue exhaustion. Metastatic lesions (Figure 1F) were microscopically confirmed in eight animals (cases 3, 5, 13, 14, 22, 25, 28 and 29) and diagnosed by ultrasound in another dog (case 23, Table 1). For IHC study, nine tissue samples with metastatic lesions from eight dogs were selected. The selection criteria were based on the amount of tissue and its conservation conditions.

### 3.2. Immunohistochemistry

#### 3.2.1. Normal Gastric Mucosa

In the normal canine gastric mucosa, a strong membranous expression of E-cad was observed, localized at the lateral cell to cell boundaries (polarized pattern), of the foveolar epithelia as well as of the deep gastric glands, either from the body or from the antrum (score 3) (Figure 2). No E-cad expression was detected in non-epithelial cells of gastric mucosa.

#### 3.2.2. Non-Neoplastic Gastric Mucosa Adjacent to Carcinomas

Non-neoplastic gastric mucosa was present in 32 of the 33 cases and in all cases E-cad expression was found similar to that observed in normal canine gastric mucosa (100%).

#### 3.2.3. Gastric Carcinomas

Abnormal expression of E-cad was observed in 29 out of 33 GC cases (87.9%); the remaining four cases (one diffuse type and three intestinal-type carcinomas) displayed normal membranous expression of E-cad (12.1%). In intestinal type carcinomas, immunoreactivity was observed in the lateral membrane (polarized pattern, Figure 3A), while in diffuse type carcinoma it was found in the whole cell membrane (non-polarized pattern, Figure 3B). This non-polarized pattern was further observed in the diffuse/isolated-cell component of two mixed carcinomas (Table 1). A marked difference in E-cad expression was observed between normal gastric tissues and carcinomas, as normal membranous expression of E-cad was mostly reduced in carcinomas when compared with normal gastric mucosa (12.1% vs. 100%; *p* < 0.0002).

In 26 cases, a heterogeneous staining (score 2) was observed, mostly characterized by areas of normal membranous staining combined with cytoplasmic and/or absence of staining (Table 1). Diffuse cytoplasmic staining was detected in more than 10% of the tumor area in 21 cases (four intestinal type and 12 diffuse type carcinomas, in both intestinal and diffuse/isolated-cell component of three mixed carcinomas and in the intestinal component of two mixed carcinoma), mainly in combination with membranous staining (Figure 3C). In most of these cases, the combined cytoplasmic and membranous staining was observed in neoplastic cells arranged in nests, whereas in isolated tumor cells the staining was only diffuse cytoplasmic. The absence of E-cad immunostaining (score 0) was observed in three cases (three diffuse type carcinomas), affecting more than 90% of the tumor area (Figure 3D), but in ten cases (one intestinal type and nine diffuse-type carcinomas), negative areas for E-cad were also noted. This abnormal pattern was mostly observed in isolated or small groups of tumor cells, usually located at the deepest tumor-invasive fronts. All mixed carcinomas displayed an overall heterogeneous staining pattern of E-cad (Table 1).

The relationship between E-cad expression and the clinicopathological features is depicted in Table 3.

Based on Lauren classification, the results showed that abnormal expression of E-cad was more frequent in diffuse and in indeterminate type carcinomas (95% and 100%, respectively) than in intestinal type carcinomas (62.5%; *p* = 0.0392). A significant association was also observed between E-cad expression and tumor differentiation, as abnormal E-cad expression was more frequent in poorly/undifferentiated carcinomas (96.0%) than in well-differentiated carcinomas (62.5%; *p* = 0.0115). There was a significant association between E-cad expression and depth of tumor invasion, as abnormal E-cad expression was more frequently observed in carcinomas invading deeper layers (muscular and serosa; 80% and 100%) than that restricted to the most superficial layer (0%; *p* = 0.0117). Abnormal expression of E-cad was significantly more frequent in GCs with neoplastic emboli (100%) than in those without (73.3%, *p* = 0.0194). However, there were no significant differences between E-cad expression and the presence of metastatic lesions (*p* = 0.1371).

E-cadherin was expressed in neoplastic emboli in 15 of the 16 (93.8%) cases evaluated; in metastases, it was present in all cases (9/9; 100%). When comparing E-cad expression in neoplastic emboli with that of the primary tumor, six cases showed normal staining pattern of E-cad (Figure 3E), unlike the abnormal pattern of the individual primary tumor; nine cases exhibited an abnormal pattern (score 2) similar to that exhibited by the primary tumor; and one case showed no E-cad expression (score 0), in contrast to the abnormal pattern (score 2) of the primary lesion. (Table 1). Regarding metastatic lesions, seven cases presented an abnormal E-cad staining pattern (score 2) similar to that of the primary lesion, while in the remaining two cases the E-cad expression was normal, in contrast to the abnormal pattern of the primary tumor (Table 1; Figure 3F).

No statistically significant differences in E-cad expression were observed on the basis of the sex, age, and weight of the canine patient’s, tumor location and histological type according to WHO classification (Table 3).

## 4. Discussion

E-cadherin plays an important role in epithelial cell–cell adhesion and in the maintenance of tissue architecture. Disorders in the expression and/or function of this glycoprotein result in loss of intercellular adhesion, with putative and consequent cell transformation and tumor progression [39]. Altered expression of E-cad has been documented in several human [27,40,41] and canine cancers [42,43,44], being related to decreased differentiation, invasiveness and/or metastasis. In the present study, E-cad immunohistochemical expression in canine GCs, as well as in NNM, neoplastic emboli, and metastatic lesions was investigated, and possible associations with the clinicopathological features of the tumors were evaluated.

In this study, the most affected breed was crossbreed (21.2%) followed by Chow-Chow (12.1%). Other breeds such as Collie, Poodle, and Belgian Shepherd also appeared in this study, although less frequently. As previously reported, the mean age of the studied dogs was 10.1 years [45] and a predominance of males was found [2,32,46]. In humans, the protective effect of estrogen has been suggested as a possible explanation for the lower risk of gastric cancer in women compared to men [47]. Nevertheless, the cause of the higher incidence of this type of lesion in male dogs remains to be elucidated. Additionally, in line with previous investigations, the most frequent clinical sign in dogs was vomiting [11,48], followed by weight loss. In addition, our data support the preferential localization of GCs in the antral region [8,49].

Previous studies using the WHO classification for domestic animals demonstrated controversial data regarding the percentage of canine GC histological types. In some studies, tubular carcinoma was the most frequent histological type [2,50] while others reported an increased frequency of undifferentiated and/or signet ring cell subtypes [9,32,51]. In the present study, using the human WHO classification, signet ring cell carcinoma was the most common histological type, followed by poorly cohesive type. Based on Lauren classification, we found a higher frequency of diffuse type carcinomas in dogs, which is in accordance with former studies [7,9,19,52].

In human gastric cancer, the intestinal type usually occurs in older males, whereas the diffuse type affects younger people and frequently females [47]. In a previous study in dogs, no statistical correlation with breed, sex, or age was found with regards to each canine GC subtype [32]. Our study also failed to find a relationship between Lauren histological type and the sex, age, or weight of the dogs. However, a significant association was found between tumor differentiation and the sex of the animals, as poorly/undifferentiated carcinomas were more frequently observed in males (89.5%) than in female dogs (57.1%; *p* = 0.0322).

In non-neoplastic canine gastric mucosa, the foveolar epithelium and deep gastric glands, either from the body or from the antrum, displayed a membranous immunoreactivity pattern of E-cad, consistent with data obtained in normal human [27,30] and canine gastric tissues [32,33]. In humans, non-neoplastic gastric mucosa adjacent to carcinomas showed normal E-cad expression [29]. Thus, our observations reflect the normal location of this intercellular adhesion molecule in the canine gastric mucosa.

Studies regarding E-cad in human GCs described a reduced or abnormal expression in about 17% to 92% of the cases [27,53,54]. Many variables could explain this wide variation range, such as tumor heterogeneity, different used antibodies, and the scoring system adopted for the immunoreactivity evaluation. In our study, the vast majority (87.9%) of canine GCs showed an abnormal expression of E-cad. Moreover, herein significant differences in E-cad expression between non- and malignant neoplastic tissues were identified (*p* < 0.0002). These findings suggest that this molecule plays a role in cell adhesion and maintenance of normal epithelial morphology of the gastric mucosa in dogs, with disruption of its function possibly being a key event in the development of canine gastric tumors. The molecular mechanisms underlying loss of normal expression of E-cad in human gastric cancer include promoter hypermethylation [55], somatic and germline mutations [56,57], activation of E-cad transcriptional repressors (e.g., Snail and Slug) [58,59], and reduced or lost expression of a few microRNAs (e.g., miR-200 family and miR-101) [60,61]. Further studies are needed to clarify which mechanism(s) are associated with abnormal expression of E-cad in canine gastric cancer.

E-cadherin interacts with ß- or γ-catenins, which in turn bind E-cad to the cytoskeleton through α-catenin [62]. Abnormal expression of α- or ß-catenin due to mutation, deletion, or post-translational modification (e.g., by phosphorylation) may account for loss of cell adhesion in the tumors retaining normal expression of E-cad [63]. In the present study, normal membranous staining of E-cad was found in three intestinal, one diffuse, and in the diffuse/isolated-cell component of two mixed carcinomas. In agreement to what has been described in human [27,31,64,65] and canine [32] GCs, in the present study a concurrent cytoplasmic subcellular location of E-cad was found in 21 of the 33 canine GC cases. Cytoplasmic localization of E-cad had already been described in other canine tumors (e.g., mammary, melanocytic) [43,66]. This subcellular localization may be due to various processes, such as: (1) endocytotic internalization of cell contact domains containing junctional cadherins [67]; (2) mutations or partial deletions of the E-cad gene, resulting in a defective protein which is not transported to the cell membrane [68]; (3) vesiculation of the Golgi apparatus, a structural change that has been described to occur in neoplastic cells [69]; and (4) disturbed polarization of the cell or disturbed interactions between E-cad and the cytoskeleton [31].

In the present investigation E-cad expression was separately evaluated in the two components of the mixed carcinomas, similarly to previous studies in human GCs [27,64]. Both human reports found abnormal E-cad expression mainly in diffuse/isolated cell-type component (17/18 and 16/16, respectively), while in the intestinal/glandular component some cases exhibited a normal membranous pattern of staining (8/18 and 3/16, respectively). In our study, normal membranous pattern of staining was found in the diffuse/isolated cell-type component of two mixed carcinomas (40%) and the intestinal/glandular component of all mixed carcinomas exhibited abnormal E-cad expression.

In a recent study in dogs, the authors found a severe reduction or complete loss of E-cad immunoreactivity in diffusetype carcinomas. However, no statistically significant associations with histological type were found [33]. Our results showed a significantly higher frequency of abnormal expression of E-cad in diffuse and indeterminate carcinomas (95.0% and 100%, respectively) than in intestinal carcinomas (62.5%; *p* = 0.0392). These results are similar to those described in other reports of human GCs [27,31]. The present study also demonstrated a significant association between the E-cad expression and tumor differentiation as the percentage of cases exhibiting abnormal expression of E-cad was significantly higher in poorly/undifferentiated carcinomas (96.0%) than in well-differentiated carcinomas (62.5%; *p* = 0.0115), in accordance with a previous report in canine colorectal carcinomas [70] and other studies in human GCs [29,54]. It has been shown that the variations in the behavior of malignant tumors are related to their degree of differentiation, as defined by morphologic criteria. Indeed, poorly differentiated carcinomas, which invade and metastasize most rapidly, are associated with a poorer prognosis [71]. Together with our results, this suggests a possible role of E-cad in the morphogenesis and biological behavior of canine gastric carcinomas, as altered expression of E-cad may lead to decrease cell-cell-adhesion, disorganization of glandular morphology and dedifferentiation, favoring tumor cell invasion and metastasis.

In the current study, abnormal expression of E-cad was significantly associated with depth of tumor invasion (*p* = 0.0117), which support its role in tumor invasiveness. Our findings concur with previous studies in human GCs [72,73] but contrast with others in which such a relationship was not found [29,31].

In human GC, the reduced or abnormal expression of E-cad has been associated with, lymph vascular invasion [30,74], lymph node involvement [30] and/or distant metastasis [29]. In this study, abnormal expression of E-cad was also significantly associated with the presence of neoplastic emboli (*p* = 0.0194), but no statistical association with metastatic disease was found, (probably due to the small number of cases with metastasis analyzed). Nevertheless, our findings suggest that loss of normal E-cad expression may be a prerequisite for neoplastic cells to detach from the primary lesion, invade vessels, and eventually to metastasize. This speculation is supported by in vitro experiments showing that the loss of E-cad in human cancer cell lines is associated with dedifferentiation and invasiveness, and that restoration of E-cad expression, by transfection and expression of cDNA, suppresses invasion in vitro [75,76,77]. Thus, E-cad may act as a tumor-suppressor protein in canine gastric carcinogenesis. Previously, Shino et al. [29] evaluated the expression of E-cad in primary and metastatic lesions from human GCs and found a high proportion of metastases having normal expression of E-cad, like the individual primary tumor (28/44). If the loss of E-cad expression is indeed associated with tumor cell detachment and vascular invasion, it is possible that tumor development in the metastatic sites might need the maintenance of E-cad expression for the reconnection and anchorage of tumor cells. In the present study, only 22.2% of metastases showed normal expression of E-cad; the remaining cases showed a heterogeneous staining pattern (77.8%) characterized by areas of normal membranous staining combined with abnormal staining areas

The prognostic significance of altered expression of E-cad in human gastric cancer has been extensively studied [29,31,38]. In a previous meta-analysis including 4383 gastric cancer patients, downregulation of E-cad was significantly associated with TNM stage, the depth of invasion, lymph node metastasis, distant metastasis, grade of differentiation, vascular invasion, histological type, and poor survival [78]. In the present study, the prognostic value in terms of survival was not assessed due to the lack of detailed information regarding follow up and patient outcome in a high proportion of cases. However, a significant association between E-cad expression and histological type according to Lauren, tumor differentiation, depth of tumor invasion, and presence of neoplastic emboli was found, suggesting that E-cad might provide prognostic information in canine GCs. Our results encourage further investigations, including a larger number of canine GC cases and clinical follow-ups to verify the usefulness of E-cad as prognostic marker in canine gastric cancer.

This study has several limitations. First, it is a retrospective analysis with a relatively limited number of cases, subjected to different sampling methods (partial vs. full-thickness biopsies), and submitted to different and not standardized clinical approaches. Second, some data about clinical information were not recorded (such as clinical signs, tumor location, metastatic status, and cause of death). Third, necropsy examination was only performed in a limited number of cases, so it is reasonable to speculate that the metastatic status may not be accurate. Fourth, no correlations were made with survival.

Nevertheless, it constitutes the first investigation where altered E-cad expression in canine GCs has been significantly associated with clinicopathological parameters suggestive of poor prognosis.

## 5. Conclusions

The present study showed an abnormal expression of E-cad in malignant gastric neoplastic lesions of dogs and that this expression was significantly associated with histological type according to Lauren, tumor differentiation, depth of tumor invasion, and presence of neoplastic emboli. These findings suggest that deregulation of E-cad may play a role in canine gastric carcinogenesis and that the disruption of tissue architecture caused by loss of normal expression of E-cad is associated with invasiveness. Moreover, E-cad might have prognostic value in canine GCs. Given the pathological and behavioral similarities between canine and human GCs, dogs may represent a potential animal model to study human GCs.

## Figures and Tables

**Figure 1 vetsci-09-00172-f001:**
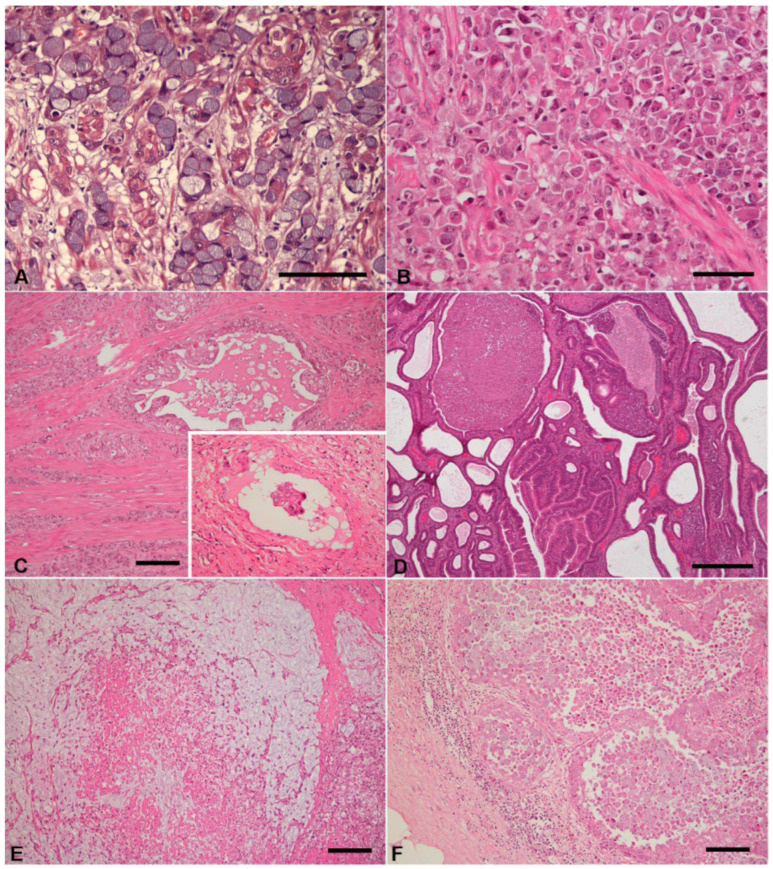
Histopathological features of gastric carcinomas and lymph node metastases. (**A**) Signet ring cell carcinoma (WHO) and diffuse type carcinoma (Lauren) composed by signet ring tumor cells (bar = 100 µm). (**B**) poorly cohesive carcinoma (WHO) and diffuse-type carcinoma (Lauren) constitute of poorly cohesive neoplastic cells (bar = 50 µm). (**C**) intestinal component of a mixed carcinoma (WHO) and indeterminate type carcinoma (Lauren) composed of a mixture of neoplastic epithelial cells organized in tubules of various sizes and nests, scattered throughout the tunica muscularis (bar = 100 µm). Inset shows neoplastic emboli (200×). (**D**) Tubular carcinoma (WHO) and intestinal type carcinoma (Lauren) consisting of numerous dilated and irregular tubular structures, occasionally with intraluminal small papillae and mucus (bar = 500 µm). (**E**) Mucinous carcinoma (WHO) and diffuse type carcinoma (Lauren) with scattered signet ring cells embedded in extracellular mucin lakes (bar = 100 µm). (**F**) Lymph node metastasis of a poorly cohesive carcinoma (WHO) and diffuse type carcinoma (Lauren), with large clusters of neoplastic epithelial cells. Note a few aggregates of lymphoid cells at the periphery (bar = 100 µm).

**Figure 2 vetsci-09-00172-f002:**
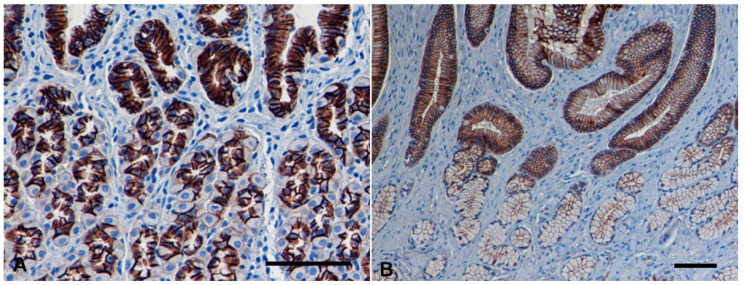
Normal canine gastric mucosa. Immunohistochemistry for E-cadherin (E-cad) counterstained with Mayer’s hematoxylin. Strong membranous expression of E-cad, at the lateral cell to cell boundaries (polarized pattern) in superficial foveolar epithelium and deep gastric glands from gastric body (**A**) and pyloric antrum (bar = 100 µm) (**B**). Note the decrease in labeling intensity in the antral glands (bar = 100 µm).

**Figure 3 vetsci-09-00172-f003:**
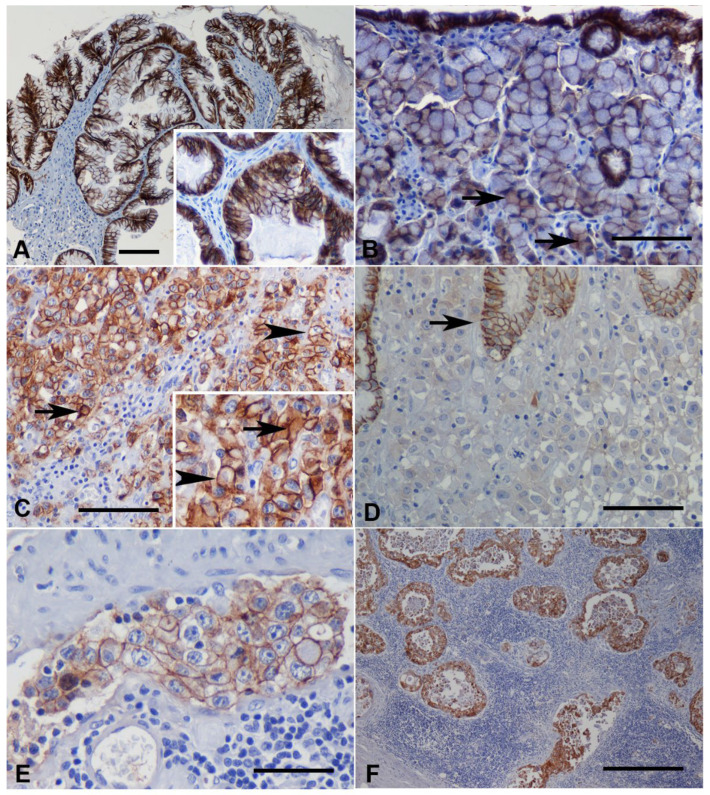
Gastric carcinomas. Immunohistochemistry for E-cad counterstained with Mayer´s hematoxylin. (**A**) Tubular carcinoma (WHO) and intestinal type carcinoma (Lauren) displaying E-cad immunoreactivity at the lateral cellular membrane (polarized pattern, score 3). (bar = 100 µm). Inset shows a higher magnification of the normal staining pattern (400×). (**B**) Signet ring cell carcinoma (WHO) and diffuse type carcinoma (Lauren) showing E-cad membranous staining, localized in the whole cell membrane (non-polarized pattern, score 3), in the majority of neoplastic cells. Cytoplasmic staining can be seen occasionally (arrow) (bar = 100 µm). (**C**) Heterogenous staining (score 2) with diffuse cytoplasmic staining (arrow) combined with membranous staining (arrowhead) in a tubular carcinoma (WHO) and intestinal type carcinoma (Lauren) (bar = 100 µm). Inset shows a higher magnification of the cytoplasmic and membranous staining (400×). (**D**) Absence of staining (score 0) with completely loss of staining in the tumor area, while in the adjacent non-neoplastic mucosa membranous staining was preserved (arrow) (bar = 100 µm). (**E**) Neoplastic emboli from a tubular carcinoma (WHO) and intestinal type carcinoma (Lauren) displaying normal pattern of immunoreactivity (bar = 50 µm). (**F**) Lymph node metastasis exhibiting heterogeneous staining (score 2), similar to that of primary tumor (tubular carcinoma (WHO) and intestinal type carcinoma (Lauren)) (bar = 500 µm).

**Table 1 vetsci-09-00172-t001:** Epidemiological data of the animals, characteristics of the tumors and main immunohistochemical findings.

Case No.	Breed	Sex/ Age (Years)	Weight, kg	Tumor Location	Histological Classification		Metastases	E-cad Immunoexpression (Score)			
					WHO Classification	Lauren		Primary Tumors		Emboli	Metastases
1	Crossbreed	F/13	≤10	Body	Tubular	Intestinal	-	N (3)		-	-
2	Labrador Retriever	F/14	26–45	Body	Tubular	Intestinal	NA	Ab (2)		-	-
3	Collie	M/11	26–45	Body and antrum	Tubular	Intestinal	Lymph node, Pancreas, Intestine *	Ab (2)		N (3)	Ab (2)
4	Miniature Poodle	F/14	≤10	Antrum	Tubular	Intestinal	-	N (3)		-	-
5	Basset Hound	F/12	26–45	Antrum	Tubular	Intestinal	Lymph node *	Ab (2)		Ab (2)	Ab (2)
6	Siberian Husky	F/12	26–45	Antrum	Tubular	Intestinal	NA	Ab (2)		Ab (2)	-
7	Siberian Husky	M/13	26–45	Antrum	Tubular	Intestinal	-	Ab (2)		-	-
8	Crossbreed (X poodle)	F/9	11–25	Antrum	Papillary	Intestinal	-	N (3)		-	-
9	Crossbreed	M/10	NR	Body	Mucinous	Diffuse	NA	Ab (2)		N (3)	-
10	Chow-Chow	M/6	26–45	Body	Mucinous	Diffuse	-	Ab (2)		Ab (2)	
11	English Bulldog	M/6	11–25	Body	Signet ring cell	Diffuse	-	Ab (2)		-	-
12	Sharpei	M/5	11-25	Body	Signet ring cell	Diffuse	-	Ab (2)		NA	-
13	Golden Retriever	M/14	26–45	Body	Signet ring cell	Diffuse	Lung *, Esophagus, Liver, Adrenal gland *	Ab (2)		Ab (0)	Ab (2)
14	Pointer	M/11	26–45	Body	Signet ring cell	Diffuse	Lymph node *	Ab (2)		Ab (2)	Ab (2)
15	Crossbreed	F/7	≤10	Body and antrum	Signet ring cell	Diffuse	-	Ab (2)		-	-
16	Cocker Spaniel	M/13	11–25	Antrum	Signet ring cell	Diffuse	-	Ab (2)		-	-
17	Chow-Chow	M/10	26–45	Antrum	Signet ring cell	Diffuse	-	Ab (0)		-	-
18	Golden Retriever	M/10	26–45	Antrum	Signet ring cell	Diffuse	-	Ab (2)		-	-
19	Boxer	M/7	26–45	Antrum	Signet ring cell	Diffuse	-	N (3)		-	-
20	West Highland White Terrier	F/13	≤10	Antrum	Signet ring cell	Diffuse	-	Ab (2)		Ab (2)	-
21	Alaska Malamute	M/6	26–45	NA	Signet ring cell	Diffuse	-	Ab (2)		-	-
22	Crossbreed	F/8	11–25	Body	Poorly cohesive	Diffuse	Intestine *, Peritoneum, Liver	Ab (2)		N (3)	N (3)
23	Crossbreed (X German Shepherd)	F/13	26–45	Body	Poorly cohesive	Diffuse	Liver	Ab (0)		-	-
24	Akita	M/9	26–45	Body	Poorly cohesive	Diffuse	NA	Ab (0)		Ab (2)	-
25	German Shepherd	M/12	26–45	Body and antrum	Poorly cohesive	Diffuse	Esophagus *, Lymph node	Ab (2)		Ab (2)	Ab (2)
26	Shih Tzu	F/10	≤10	Antrum	Poorly cohesive	Diffuse	-	Ab (2)		NA	-
27	Chow-Chow	M/9	26–45	Antrum	Poorly cohesive	Diffuse	-	Ab (2)		-	-
28	Crossbreed	F/7	NR	Antrum	Poorly cohesive	Diffuse	Intestine *	Ab (2)		-	N (3)
								Intestinal component	Diffuse component		
29	Belgian Shepherd	F/11	11–25	Body	Mixed	Indeterminate	Lymph node *	Ab (2)	Ab (2)	Ab (2)	Ab (2)
30	Collie	M/11	26–45	Body	Mixed	Indeterminate	-	Ab (2)	N (3)	N (3)	-
31	Chow-Chow	F/11	26–45	Antrum	Mixed	Indeterminate	-	Ab (2)	Ab (2)	N (3)	-
32	Labrador Retriever	M/8	26–45	Antrum	Mixed	Indeterminate	NA	Ab (2)	N (3)	Ab (2)	-
33	Standard Poodle	M/8	11–25	Antrum	Mixed	Indeterminate	-	Ab (2)	Ab (2)	N (3)	-

M—male; F—female; NR—not recorded; NA—not available; N—normal; Ab—abnormal; * Cases submitted to IHC evaluation. In case 13, E-cad expression in both lung and adrenal metastases was score as 2.

**Table 2 vetsci-09-00172-t002:** Association between histological type according Lauren classification and tumor differentiation and sex, age, and weight of the dogs.

	No. of Cases	Histological Diagnosis			*p*-Value	Tumor Differentiation		*p*-Value
		Intestinal	Diffuse	Indeterminate		Well- Differentiated	Poorly/ Undifferentiated	
Sex								
Male	19	2	14	3	0.0929	2	17	**0.0322**
Female	14	6	6	2		6	8	
Age, years								
<10	13	1	10	2	0.1858	1	12	0.0737
≥10	20	7	10	3		7	13	
Weight, kg								
≤10	5	2	3	0	0.6924	2	3	0.6023
11–25	7	1	4	2		1	6	
26–45	19	5	11	3		5	14	

**Table 3 vetsci-09-00172-t003:** Relationship between E-cad expression in 33 canine gastric carcinomas and clinicopathological parameters.

Clinicopathological Parameters	No. of Cases	E-cad Immunoexpression		*p*-Value
		Normal (n = 4)	Abnormal (n = 29)	
		n (%)	n (%)	
**Sex**				
Male	19	1 (5.3%)	18 (94.7%)	0.1597
Female	14	3 (21.4%)	11 (78.6%)	
**Age, years**				
<10	13	2 (15,4%)	11 (84.6%)	0.6433
≥10	20	2 (10.0%)	18 (90.0%)	
**Weight, kg ^a^**				
≤10	5	2 (40%)	3 (60%)	0.1185
11–25	7	1 (14.3%)	6 (85.7%)	
26–45	19	1 (5.3%)	18 (94.7%)	
**Tumor location ^b^**				
Antrum	16	3 (18.8%)	13 (81.3%)	0.5287
Body	13	1 (7.7%)	12 (92.3%)	
Body and antrum	3	0 (0%)	3 (100%)	
**Histological diagnosis**				
WHO classification				
Tubular	7	2 (28.6%)	5 (71.4%)	0.0503
Papillary	1	1 (100%)	0 (0%)	
Mucinous	2	0 (0%)	2 (100%)	
Signet ring cell	11	1 (9.1%)	10 (90.9%)	
Poorly cohesive	7	0 (0%)	7 (100%)	
Mixed	5	0 (0%)	5 (100%)	
Lauren				
Intestinal	8	3 (37.5%)	5 (62.5%)	**0.0392**
Diffuse	20	1 (5.0%)	19 (95.0%)	
Indeterminate	5	0 (0%)	5 (100%)	
**Tumor differentiation**				
Well-differentiated	8	3 (37.5%)	5 (62.5%)	**0.0115**
Poorly/undifferentiated	25	1 (4.0%)	24 (96.0%)	
**Depth of tumor invasion ^c^**				
Mucosa	1	1 (100%)	0 (0%)	
Muscular	10	2 (20.0%)	8 (80.0%)	**0.0117**
Serosa	12	0 (0%)	12 (100%)	
**Neoplastic emboli**				
Present	18	0 (0%)	18 (100%)	**0.0194**
Absent	15	4 (26.7%)	11 (73.3%)	
**Metastatic lesions ^d^**				
Present	9	0 (0%)	9 (100%)	0.1371
Absent	19	4 (21.1%)	15 (78.9%)	

^a^ Weight was not recorded in two cases; ^b^ tumor location was impossible to obtain in one case; ^c^ for statistical analysis only full-thickness biopsies were included; ^d^ information regarding metastatic lesion was impossible to obtain in five cases.

## Data Availability

The data presented in this study are contained within the manuscript.

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
