# Peer review of "E-cadherin Expression in Canine Gastric Carcinomas: Association with Clinicopathological Parameters"

_vetsci, 2022, doi:10.3390/vetsci9040172_

Round 1
Reviewer 1 Report
This original study describes the immunohistochemical "normal" vs. "abnormal" expression of E-cadherin in canine gastric carcinoma. E-cadherin is a relevant tumor marker associated with local invasion and is therefore generally of great interest, however, this study provides relatively few new information. The most important limitation is the used scoring system for E-cadherin expression (see below), which destinguished the gastric carcinoma in 3 vs. 33 cases. This low group size (N = 3) allows only very limited power for the statistical analysis, even when the authors report "siginficant" results. The manuscript is well-structured and the illustration can be improved (see below). Specific comments are listed below.
LIne 70 and 137: The Lauren classification system is not well known among veterinary pathologists. The authors should provide a description of this system and explain the meaning for canine tumors. Has this system been correlated with clinical outcome in previous studies?
Line 142: Which criteria was used to diagnose lymphovascular invasion? The reviewers refer the authors to the recent guidelines by the VCGP group: https://www.vcgp.org/ or https://cloud.cldavis.org/index.php/s/7PwYdTj8ST3gGND
Section 2.5: As mentioned above, the reviewer fells that the used scoring system for E-cadherin expression is the biggest limitation of the study. The score categories are very confusing to me. The score categories need to be better explained and ideally exemplary images of each score should be provided. Please explain the rationale of the categories of the semi-quantitative E-cadherin score. They do not seem to be consecutive scores of increasing/decreasing degree of abnormal expression (as a semi-quantitiative score would suggest) as some scores categories are defined based on the labeling intensity and other score categories based on the labeling localisation/pattern. How does score no. 2 ("complete membranous cell labeling") differ from normal (score no. 5)? For statistical analysis of this study, score category 0-4 were combined to "abnormal", which introduced the problem that >90% of the cases are on one category, which limits statistical comparison of the groups. Even though the authors report "significant" p-values, I have doubts. I wonder if a contigious scoring system with decreasing/increasing label intesity (or based on other criterion) would allow a better comparison.
Figures: FIgure 1F and 1E do not show lymphatic vessels with emboli and ultrasound findings as suggested in the text. I recommend to use images with higher resolution in order to appreciate the IHC labeling pattern.
Discussion: The discussion mostly repeats the findings of the study. Teh relevance of the main findings should be highlighted and limitaitons of the study need to be adressed. Also, discussion on the diagnostic / prognostic use of this marker in gastric carcinoma would be interesting
Author Response
This original study describes the immunohistochemical "normal" vs. "abnormal" expression of E-cadherin in canine gastric carcinoma. E-cadherin is a relevant tumor marker associated with local invasion and is therefore generally of great interest, however, this study provides relatively few new information. The most important limitation is the used scoring system for E-cadherin expression (see below), which distinguished the gastric carcinoma in 3 vs. 33 cases. This low group size (N = 3) allows only very limited power for the statistical analysis, even when the authors report "significant" results. The manuscript is well-structured and the illustration can be improved (see below). Specific comments are listed below.
Line 70 and 137: The Lauren classification system is not well known among veterinary pathologists. The authors should provide a description of this system and explain the meaning for canine tumors. Has this system been correlated with clinical outcome in previous studies?
In accordance with the reviewer´s suggestion, a description of the Lauren classification system and its meaning for canine tumors was provided in lines 154-159 of the materials and methods section of the revised manuscript.
To our knowledge, and although this scheme has been successfully adapted to the dog, a correlation between histological type according to Lauren and prognosis has not yet been demonstrated in canine GCs. This information was mentioned in lines 73-79 of the revised manuscript.
Line 142: Which criteria was used to diagnose lymphovascular invasion? The reviewers refer the authors to the recent guidelines by the VCGP group: https://www.vcgp.org/ or https://cloud.cldavis.org/index.php/s/7PwYdTj8ST3gGND
The presence of neoplastic emboli was considered whenever the criteria 2 or 3 proposed by Moore et al. (2021) were verified. This information was mentioned in lines 167-170 of the revised manuscript, please check.
Section 2.5: As mentioned above, the reviewer fells that the used scoring system for E-cadherin expression is the biggest limitation of the study. The score categories are very confusing to me. The score categories need to be better explained and ideally exemplary images of each score should be provided. Please explain the rationale of the categories of the semi-quantitative E-cadherin score. They do not seem to be consecutive scores of increasing/decreasing degree of abnormal expression (as a semi-quantitiative score would suggest) as some scores categories are defined based on the labeling intensity and other score categories based on the labeling localisation/pattern. How does score no. 2 ("complete membranous cell labeling") differ from normal (score no. 5)? For statistical analysis of this study, score category 0-4 were combined to "abnormal", which introduced the problem that >90% of the cases are on one category, which limits statistical comparison of the groups. Even though the authors report "significant" p-values, I have doubts. I wonder if a contigious scoring system with decreasing/increasing label intensity (or based on other criterion) would allow a better comparison.
The authors understand the reviewer´s concern. Thus, the scoring system for E-cadherin expression was replaced by that commonly used in human gastric cancer studies (Jawhari et al., 1997; Machado et al., 1998). In this context, the abstract, materials and methods, results and discussion section have undergone major changes.
Figures: Figure 1F and 1E do not show lymphatic vessels with emboli and ultrasound findings as suggested in the text. I recommend to use images with higher resolution in order to appreciate the IHC labeling pattern.
The reviewer is right. The location of figures in the main text has been changed (line 249 and line 281). In addition, images with higher resolution have been included.
Discussion: The discussion mostly repeats the findings of the study. The relevance of the main findings should be highlighted, and limitations of the study need to be addressed. Also, discussion on the diagnostic / prognostic use of this marker in gastric carcinoma would be interesting.
The discussion has undergone major changes. Repeated findings were deleted, main findings were highlighted, and limitations of the study were added (please check lines 598-604). Additionally, as suggested by the reviewer, the prognostic value of this marker has been discussed in lines 585-597.
Reviewer 2 Report
Thank you for sending me the research article paper “E-cadherin expression in canine gastric carcinomas: association with clinicopathological parameters” for review in the Cancers. In the article of Flores et al., the author discussed the role of E-cadherin in the development of gastric cancer in the canine. However, there are important points that should be improved.
- Author should reduce the length of the introduction part. It looks a little lengthy. It would be good to combine some paragraphs.
- Figure-1, 2 and 3 need to be presented in a way that is more scientific. Author should highlight the important changes with an arrow. Also there is a need to provide the same brightness and color of the figure.
- Author should provide the higher magnification figures along with lower magnification.
- Conclusion is too long. Should write with more precision.
- There is a need to cross the level of E-cadherin with other types of experiments. e.g. qPCR, western blotting or ELISA.
Author Response
Thank you for sending me the research article paper “E-cadherin expression in canine gastric carcinomas: association with clinicopathological parameters” for review in the Cancers. In the article of Flores et al., the author discussed the role of E-cadherin in the development of gastric cancer in the canine. However, there are important points that should be improved.
- Author should reduce the length of the introduction part. It looks a little lengthy. It would be good to combine some paragraphs.
As suggested by the reviewer, the introduction has been shortened.
- Figure-1, 2 and 3 need to be presented in a way that is more scientific. Author should highlight the important changes with an arrow. Also there is a need to provide the same brightness and color of the figure.
As suggested by both reviewers, images presentation was changed. The important changes have been highlighted with arrow and brightness and colour have been added to images.
- Author should provide the higher magnification figures along with lower magnification.
As suggested by the reviewer, images with higher resolution have been added. Please check.
- Conclusion is too long. Should write with more precision.
Since the manuscript was submitted to major changes in all sections, including the conclusion which as, as proposed, rewritten and shorten. Please check.
- There is a need to cross the level of E-cadherin with other types of experiments. e.g. qPCR, western blotting or ELISA.
Since the time provided for the revision was not enough for the authors to perform these experiments, the authors resorted to the existing literature to compare these findings with others resulting from the application of different techniques.
Round 2
Reviewer 1 Report
The revised version of this manuscript has significantly improved. The authors have acknowledged limitations that cannot easily be avoided for their study. I only have few additional comments:
1) Table 1: Please explain/describe why the IHC pattern for some, but not all tumors, was distuingished into interstitial and diffuse component
2) I strongly recommend to put back the actual P-values in Table 2 and 3 instead of "NS". I know this is sometimes done, but it does not allow appropriate interpretation of the results. This goes back to the discussion "what is significant and what is not significant", but there is a big difference between 0.06 ("almost significant") vs. 0.6, for example, which are now all summarized under the term "not significant". If the actual p-value is provided the reader can interpret the results much better.
Author Response
# REVIEWER 1
The revised version of this manuscript has significantly improved. The authors have acknowledged limitations that cannot easily be avoided for their study. I only have few additional comments:
1) Table 1: Please explain/describe why the IHC pattern for some, but not all tumors, was distinguished into interstitial and diffuse component.
The authors believe that the reviewer is referring to "intestinal" instead of "interstitial component". In addition to the histological types described in the WHO classification for domestic animals, the human WHO classification includes the mixed and poorly cohesive histological types. Morphologically, mixed carcinomas are characterized by displaying, in the same tumour, an intestinal/glandular component and a diffuse/isolated-cell type component. In a previous study, Machado et al. (1998a) demonstrated a significant association between abnormal E-cad expression and the diffuse histologic type. This association was maintained in mixed carcinomas where E-cad expression was evaluated separately in the two components; indeed, abnormal E-cad expression in the diffuse/isolated-cell type component was significantly higher than in the intestinal/glandular component. According to these authors, the significant relationship between an abnormal expression of E-cad and diffuse/isolated-cell type carcinomas, which is kept both in "pure" carcinomas and in the diffuse component of mixed carcinomas, is in line with the hypothesis that E-cad plays a crucial role in the adhesion among neoplastic cells, which, if disrupted, leads to dissociation of cells and scattered growth. This hypothesis was supported by another study in which the same authors detected mutations in the E-cad gene in 70% of diffuse carcinomas and in the diffuse component/isolated-cell type of 5/6 mixed carcinomas, suggesting the existence of genotypically divergent tumour clones in mixed carcinomas (Machado et al., 1998b). Based on these results the authors concluded that E-cad inactivation is significantly related to the diffuse type, not only in “pure” diffuse carcinomas but also in the diffuse component of mixed tumors.
As some of the canine gastric carcinomas cases included in the present study fit the description of mixed carcinoma, the expression of E-cad was evaluated separately in both components of in order to verify a putative association between the expression of E-cad and the different histological components, as described for human gastric carcinomas. Similar to human studies, we found a significant association between abnormal expression of E-cad and the diffuse histological type. However, in mixed carcinomas, 2/5 had normal pattern in the diffuse/isolated-cell type component and abnormal pattern in the intestinal/glandular component and 3/5 had abnormal pattern in both components.
2) I strongly recommend to put back the actual P-values in Table 2 and 3 instead of "NS". I know this is sometimes done, but it does not allow appropriate interpretation of the results. This goes back to the discussion "what is significant and what is not significant", but there is a big difference between 0.06 ("almost significant") vs. 0.6, for example, which are now all summarized under the term "not significant". If the actual p-value is provided the reader can interpret the results much better.
In accordance with the reviewer´s suggestion, P-values were added. Please check.